# Unusual Localization of AIDS-Related Kaposi’s Sarcoma in a Heterosexual Male during the COVID-19 Pandemic: A Case Report

**DOI:** 10.3390/tropicalmed9020047

**Published:** 2024-02-13

**Authors:** Manuela Arbune, Monica-Daniela Padurariu-Covit, Carmen Tiutiuca, Raul Mihailov, Elena Niculet, Anca-Adriana Arbune, Alin-Laurentiu Tatu

**Affiliations:** 1Medical Clinic Department, Dunarea de Jos University, 800008 Galati, Romania; manuela.arbune@ugal.ro (M.A.); alin.tatu@ugal.ro (A.-L.T.); 2Clinical Hospital for Infectious Diseases, 800179 Galati, Romania; 3Doctoral School of Biomedical Sciences, Dunarea de Jos University, 800008 Galati, Romania; 4Hematology Department, Sf. Apostol Andrei Emergency County Hospital, 800578 Galati, Romania; 5Surgery Clinic Department, Dunarea de Jos University, 800578 Galati, Romania; carmen.tiutiuca@ugal.ro (C.T.); raul.mihailov@ugal.ro (R.M.); 6Ophthalmology Department, Sf. Apostol Andrei Emergency County Hospital, 800008 Galati, Romania; 7General Surgery Department, Sf. Apostol Andrei Emergency County Hospital, 800578 Galati, Romania; 8Morphological and Functional Sciences Department, Dunarea de Jos University, 800008 Galati, Romania; helena_badiu@yahoo.com; 9Pathology Department, Sf. Apostol Andrei Emergency County Hospital, 800578 Galati, Romania; 10Multidisciplinary Integrated Center for Dermatological Interface Research, 800010 Galati, Romania; anca.arbune@icfundeni.ro; 11Neurology Department, Fundeni Clinical Institute, 077086 Bucharest, Romania

**Keywords:** AIDS-related Kaposi’s sarcoma, conjunctival Kaposi’s sarcoma, co-infections, human herpes virus-8, COVID-19, vulnerable, adherence

## Abstract

Kaposi’s sarcoma is an AIDS-defining illness and remains the most frequent tumor arising in HIV-infected patients with multifactorial etiology. We present a case of a 30-year-old Caucasian male with an 18-year history of HIV infection. The patient was presented with a one-week history of fever, non-productive cough, and skin lesions. There was an associated weakness and weight loss in a duration of 6 months. Clinical examination showed fever, generalized lymphadenopathy, lower limb edema, ascites, and violaceous cutaneous eruption comprising patches, plaques, and nodules. He also had a red nodule on the left conjunctiva, as well as on his oral mucosa. His CD_4+_ count was below 10/mm^3^ and ARN-HIV viral load was above 100,000 c/mL, in relation to the antiretroviral failure after five drug regimens. The role of co-infections in oncogenesis and the course of Kaposi’s sarcoma were considered in recent studies. Delayed diagnosis of Kaposi’s sarcoma in the present case resulted in a negative impact for this patient during the COVID-19 pandemic.

## 1. Introduction

The Kaposi’s sarcoma (KS) is a vascular tumor with endothelial cell origin and is poorly differentiated, which can affect the skin, mucous membranes, or visceral organs. The histological features of KS are increased angiogenesis, the presence of spindle-shaped cells, and erythrocyte extravasation [1].

KS develops as a multifocal tumor, usually at the mucocutaneous level, especially with locations in the extremities of the lower limbs, the face, the trunk, the genital organs, and the mucosa of the oropharynx, but also in the lymph nodes and visceral organs, with more frequent involvement of the respiratory and gastrointestinal tract. The unusual localizations of KS are the musculoskeletal system, the central and peripheral nervous systems, the larynx, the eye, the major salivary glands, endocrine organs, the heart, the thoracic duct, the urinary system, and the breast [2].

From the first description of KS a century and a half ago until now, five clinical epidemiological forms have been identified: “classic” KS, in elderly men of Mediterranean or Ashkenazi origin; African SK (endemic); KS associated with acquired immunodeficiency syndrome (AIDS) (epidemic); transplant-associated KS (iatrogenic); and non-epidemic KS (MSM but who are HIV-seronegative without any type of immunodeficiency). The most aggressive form is AIDS-related KS, mostly affecting men who have sex with men (MSM) [2,3,4].

AIDS-related KS is one of the cancer indicators for AIDS in people with CD_4_ lymphocyte counts below 200/mm^3^, estimating that 30% of people who do not receive antiretroviral treatment will develop KS [3,5].

Past research on finding an etiological agent of KS identified Kaposi’s sarcoma-associated herpesvirus (KSHV) in 1994—currently also called human herpes virus-8 (HHV8).

Nowadays, it is considered that KSHV is the cause of all cases of KS. However, even though KSHV is necessary for KS development, it is considered not sufficient to result in KS, thereby requiring other factors such as immune suppression [6,7,8,9].

## 2. Case Report

A 30-year-old Caucasian male patient, living in a rural area, married, with a confirmed diagnostic of HIV infection since the age of 12, has been presented with fever, dry cough, and skin lesions, with onset of a week before. Moreover, weakness and weight loss were progressively noted in the last six months. His spouse’s HIV status was also positive since his early childhood.

On clinical examination, the patient had a temperature of 38.6 °C; BP of 92/63 mm Hg; HR of 104/min; RR of 28/min; a Glasgow coma score of 15; edema in the lower limbs and perineum; moderate ascites; cervical, axillary, and inguinal lymphadenopathy; a diffuse left axillary violaceous patch extending to the anterior side of the chest; as well as purple-brown plaques and nodules, which were slightly elevated, of firm consistency, with irregular but well-defined margins and variable sizes, ranging from 5 to 20 mm, distributed on the face, limbs, and chest (Figure 1).

A nodular, red, vascularized 10 × 15 mm lesion was found on the right conjunctiva (Figure 2).

Similar lesions, which bled on touch, were described on the oral mucosa, mainly on the gums and palate (Figure 3).

His past medical history revealed that the patient probably acquired HIV as an infant in hospital, through unsafe injections. The HIV diagnosis was made in the context of growth and development deficiency, along with diarrhea and recurrent skin and respiratory infections.

The HIV stage at the time of diagnosis, according to Centers for Disease Control and Prevention (CDC) classification under the age of 13, was AIDS-C3. Baseline CD_4_ lymphocyte count was 71/mm^3^. The first-line antiretroviral treatment combined zidovudine, lamivudine, and lopinavir/ritonavir, with sustained viral suppression for 6 years and increased immunity to an 808/mm^3^ count of CD_4_ lymphocytes. However, after the age of 18, he practiced risky behavior, such as smoking and alcohol use, as well as non-compliance with antiretroviral therapy, with viral and immunological failure (Figure A1; Table A1).

From 2019 to 2020, continuing non-adherence to antiretroviral therapy deepened the viral and immunological failure and he presented with four episodes of febrile neutropenia associated with staphylococcal infections of the skin and soft tissues, which were resolved with surgical drainage and systemic antibiotic treatments. One of these episodes, in June 2020, was associated with a mild form of COVID-19 while undertaking care at home. In December 2020, one month before the current presentation, multiple disseminated boils, including a periarticular region of the right knee and hip, complicated with sepsis, were reported (Figure A1).

His laboratory tests revealed severe anemia, thrombocytopenia, inflammation, and hypoalbuminemia (Table A1). 

The immune status during the therapy with tenofovir/emtricitabine and rilpivirine indicates a CD_4_ lymphocyte count of 2/mm^3^, requiring pneumocystis prophylaxis with trimethoprim sulfamethoxazole, and atypical mycobacteria with clarithromycin. His CD_4_/CD_8_ ratio was 0.46, predicting poor prognosis.

A serologic test for HHV8 was not available.

A computed tomography (CT) examination of the thorax revealed bilateral pleurisy, hilar adenopathy, and multiple nodular images moderately iodophilic hyperdense disseminated in both lung areas in the. The imaging of the abdominal area showed splenomegaly, ascites, periaortic, and retroperitoneal adenopathy, and the largest lymph node was 66/13 mm. The differential diagnoses comprised Kaposi’s sarcoma, lymphoma, squamous cell carcinoma or basal cell carcinoma, but we should also consider multicentric Castleman disease. The oral mucosa biopsy showed histopathologic features of Kaposi’s sarcoma (Figure 4).

The general state worsened in the next week after hospital admission, with the extension of the skin lesions to the gums, the palate, and the conjunctiva of the left eye, associated with cachexia, anasarca, and respiratory dysfunction. The clinical, imaging, laboratory, and pathological examination data supported the final diagnosis of systemic Kaposi’s sarcoma (mucosal, cutaneous, ganglion, and visceral involvement), AIDS-C3 stage, pancytopenia, and hypoalbuminemia. Unfortunately, the immunohistochemistry examination was not available. The very rapid course, the poor physical, mental, and functional state did not give room for oncological treatment. He died less than a month after the appearance of Kaposi’s lesions, because of digestive hemorrhage, indicating the probable dissemination of the lesions at the level of the gastric mucosa. The family refused the necropsy examination.

## 3. Discussion

### 3.1. Demographic Features

The risk of KS in people with AIDS is 30–80 times higher than in the general population, although the incidence has decreased in recent years, as effective antiretroviral therapy is used and persistent viral HIV replication is suppressed [10,11].

The most affected age group is 20–59 years old, in which 95% of the reported cases are found [12].

The demographic features of the presented case are consistent with the prevalence data regarding the age and sex of AIDS-related KS. The patient is heterosexual, not reporting extramarital sexual relationships, but has developed an aggressive form.

The incidence of KS in people with HIV infection was estimated by a meta-analysis of 71 studies, at 48.154 per 100,000 person-years, with variations between different populations, being higher in MSM and lower in women [13]. A study carried out between 2001 and 2021 in the center where the presented case was followed up calculated a yearly incidence of AIDS-related KS of 17 per 100,000 people, showing the relative rarity of this form of cancer [14,15]. 

The prevalence of HHV8 in the European region is low, but the situation in Romania is unknown.

### 3.2. Epidemiologic Features of HIV Infection

Although it is the case of a young adult, the HIV infection had a long course of over 20 years. The infection probably occurred in a nosocomial manner, in the first year of life. The patient’s mother and siblings are HIV seronegative. The patient belonged to a special pediatric cohort, known in Romania, through parenteral HIV infection of a large number of infants in 1988–1990, especially with F viral subtype [16,17,18]. He was a late presenter because the HIV diagnosis was late. Although the initial response to antiretroviral therapy was favorable, the patient became non-compliant to the treatment, resulting a gradual viro-immunological decline, associated with numerous opportunistic and intercurrent infections. Non-compliance is the consequence of multifactorial determinants, including therapeutic fatigue after long-term daily medication, neurocognitive dysfunction due to sequelae of tuberculous meningoencephalitis or HIV neuro impairment, personality traits involving low self-esteem, a history of low-level education, and risk behaviors such as alcohol use and deficient in social integration and support.

### 3.3. KS Ocular Involvement

The patient developed a high-risk systemic form of the disease, evaluated according to the severity criteria proposed by the management guidelines for KS: an association of edemas, extensions of oral and visceral lesions, and immunosuppression with a CD_4_ lymphocyte count below 200/mm^3^ [11].

The visceral extension of KS as in the presented case is reported in the medical literature in 15% of AIDS patients, often associated with tumoral edema and with a poor prognosis [19]. 

Ophthalmic localization of KS is rare. Until the beginning of the AIDS pandemic, only 25 cases of ophthalmic KS were reported in the US [11]. 

In AIDS patients, ophthalmic involvement was reported in 20–24% of cases in the US, with the first signs of the disease found in 4–12% of patients. Most of the eye involvement concern the conjunctiva (10–75%) or the eyelids (25–80%). The ocular caruncle and lacrimal sac are rarely affected. Symptoms compatible with the ocular localization of KS can be pain, photophobia, blurred vision, epiphora, foreign body sensation, or dry eyes. The clinical manifestations of ocular KS may not be typical, with the appearance of red eyes or conjunctival hemorrhages. The suggestive appearance of KS lesions is a macule, plaque or nodule, in bright red, purple or brown color, highly vascularized, and usually with telangiectatic vessels [20,21,22]. 

The expansion of the eye tumor can lead to severe damage of the eye, its appendages, the orbit, or even the carotid. Involvement of the eyelids causes functional and aesthetic damage. Lagophthalmos and trichiasis can cause deep irritation, infections, and corneal scarring [2]. 

The differential diagnosis in the case of early lesions can be made with subconjunctival hemorrhage, pyogenic granuloma, conjunctival cyst, and pinguecula, while larger lesions can be confused with cavernous hemangioma, hemangiopericytoma, conjunctival lymphangioma, lymphoma, malignant melanoma, squamous cell carcinoma or metastases [22,23,24].

### 3.4. Co-Infection Features and Hypotheses of the Pathogenic Role of KS

#### 3.4.1. Human Herpes Virus-8

In our case, viral level testing for HHV8 was not available, but it is demonstrated that KS cases are associated with this infection. The life cycle of HHV8 has a latent replication phase and a lytic phase. During the latent phase, the expression of genes associated with HHV8 is limited, but epigenetic factors can switch between the latent and the lytic form. Reactivation of the latent form can be caused by environmental factors, chemical agents, or hydrogen peroxide from reactive oxygen species [11,12].

During the viral replicative phase, IL6 secretion increases, which can induce vascular endothelial growth factor and neoplastic proliferation [25]. Infections with HHV8 viruses are associated with a disturbance of the regulation of the expression of oncogenic and onco-suppressive genes, favored by immunosuppression and cellular activation during the evolution of HIV [9,26].

The lifelong persistence of HHV8 is achieved by the modulation of host cell signaling, in which several viral proteins of HHV8 participate in the response of the host’s immune system. This fact is reflected by the higher incidence of HHV8 infection in immunocompromised people [7,8].

The humoral immune response has been demonstrated by epidemiological studies, which identified antibodies against viral proteins, the most immunogenic being latency-associated nuclear antigen (LANA) in the latent phase and K8.1 in the lytic phase. However, anti-HHV8 antibodies are rarely neutralizing [8].

The cellular immune response has been evaluated by recent studies, which have found variable individual responses of CD_4_ and CD_8_ lymphocytes to a wide variety of viral antigens [9].

The variation in anti-/proinflammatory cytokines reflects both the course of HIV and HHV8 infection. The evaluation of serum concentrations of IL-6, TNF-a, IL-10, and CXCL10 in different groups of people revealed significantly higher serum concentrations of IL10, IL6, and CXCL10 in patients with AIDS-related KS compared to those coinfected with HIV/HHV8, as well as in those with visceral forms of KS compared to the cutaneous forms [26,27,28]. The concentration of TNF-α was significantly higher in the group with HIV mono-infection compared to the group with HIV/HHV8 co-infections. These data indicate the complex interaction between HIV, HHV8, and immunity [28].

#### 3.4.2. *Staphylococcus aureus*

*Staphylococcus aureus* (*S. aureus*) was isolated in recurrent episodes of infections in our patient, and is frequently reported in patients with HIV. The global emergence of methicillin-resistant *S. aureus* (MRSA) was accompanied by an increase in the colonization and prevalence of infections in HIV patients [29].

Experimental studies have demonstrated that pathogen-associated molecular patterns (PAMPs) specific to *Staphylococcus aureus*, can also induce the reactivation of HHV8 in the lytic stage at the level of oral epithelial cells, favoring the activation of the expression of oncogenic genes. On the other hand, HHV8 infection increases the early internalization of *S. aureus* in the tumor tissue, possibly promoting the formation of new KS lesions. These findings suggest the possible benefits of anti-staphylococcal prophylaxis for the treatment of AIDS-associated KS [30].

#### 3.4.3. SARS-CoV-2

A mild form of COVID-19 in a severe immunocompromised patient was noted, but the postinfectious outcome was unpredictable. The symptomatology of COVID-19 is variable in terms of spectrum and intensity and can be ignored when it overlaps with the usual manifestations of patients with chronic diseases [31].

It is relevant to mention that neither vaccination nor anti-spike serum antibodies were available tools in our center for that moment of case management.

As already described for other viral co-infections, SARS-CoV-2 can produce the transition of HHV8 from the latent phase to the lytic replication phase by stimulating the release of pro-inflammatory cytokines, leading to severe disease and death [32,33,34,35].

Oxidative stress in patients with SARS-CoV 2 infection is exacerbated by co-infections, associated with the excess production of oxygen-free radicals and defects in the antioxidant system. Hypoxia during COVID-19 can produce H_2_O_2_ superoxide, which upregulates the expression of inflammatory cytokines, increasing oxidative stress and thereby activating macrophages, neutrophils, and endothelial cells [36,37]. COVID-19 co-infections have the potential to promote virus-associated cancers [38,39].

#### 3.4.4. Genetic Factors of Susceptibility to Infections

Although no genetic tests were performed in the presented case, the individual characteristics of the host played a role in the causation of the severity and rapid course of KS.

The significance of some genetic factors of the host, such as the *MBL2* gene polymorphism, has been studied in the case of HHV8, but also for other viral infections, such as human papilloma virus, hepatitis B and C viruses, herpes simplex virus 2, Epstein–Barr virus, and SARS-CoV-2 [40,41,42,43,44].

Mannose binding lectin 2 (MBL2) is a gene on chromosome 10q11.2 that encodes serum proteins secreted by hepatocytes, involved in the extracellular pathogenic microbes’ opsonization and phagocytoses. The MBL2 polymorphism gene causes defects in the polymerization of MBL proteins, which have functional deficiencies or low serum levels, associated with high risk or recurrence of infections [45].

Studies on the *MBL2* gene polymorphism in HIV-seropositive people did not show an association with HHV8 infection but reported an increased susceptibility to HIV infection in the population of European origin, in relation to the LX/LX genotype. The interpretation of these results must be conducted with caution because the available studies are limited and do not agree with data from other populations in Africa or America [46,47].

According to the *MBL2* gene, three haplotypes were highlighted, with increased, intermediate, or low expression. The intermediate haplotype was correlated with HIV/HHV8 co-infection, compared to HIV mono-infected individuals. In addition, no patient who developed KS had a haplotype with increased expression, with the intermediate haplotype identified in all patients. These results suggest that the intermediate haplotype may interfere with the clinical development of KS in patients with HIV/HHV8 co-infection [48].

#### 3.4.5. Possible Co-Existence of Multicentric Castleman Disease and Kaposi’s Sarcoma

Considering the ascites and bilateral pleurisy in our case, pleural biopsy should have been performed to differentiate opportunistic infections or other malignancies [49]. A similar presentation as our patient, with fever, weight loss, generalized lymphadenopathy, and splenomegaly is noted in multicentric Castleman disease [MCD], but additional dermatologic violaceous plaques and nodules were found in the present case. Furthermore, MCD has been reported to be associated with HHV8 and HIV infections, usually in adults, before the age of 30 years [50,51,52]. Kaposi’s sarcoma inflammatory cytokine syndrome (KICS) is recently described, characterized by a high HHV8 viral load and increased levels of IL-10 and IL-6. The diagnosis of KICS is done by the exclusion of multicentric Castleman disease and serious recurrent infections [52]. The possible association of MCD is only speculative, as the limited access to non-COVID interventions, severe conditions, and the rapid progression to death did not allow for more investigations.

## 4. Conclusions

KS continues to be a threat to HIV-infected patients with severe immunosuppression, despite the availability of active antiretroviral therapy. The non-compliance to antiretroviral therapy and the progression of immunodepression increased the risk of co-infections, with the role of co-pathogens in the oncogenesis of Kaposi’s sarcoma. Severe and prolonged immunosuppression is associated with systemic, severe, and rapidly progressive forms of Kaposi’s sarcoma associated with AIDS. Ocular localization of Kaposi’s sarcoma is a rare form, associated with poor prognosis. COVID-19 had a negative impact on patients with HIV/AIDS infection through the potentially favored role of oncogenesis, as well as through the administrative impact of limiting access to hospital medical services and delaying the diagnosis of serious diseases.

## Figures and Tables

**Figure 1 tropicalmed-09-00047-f001:**
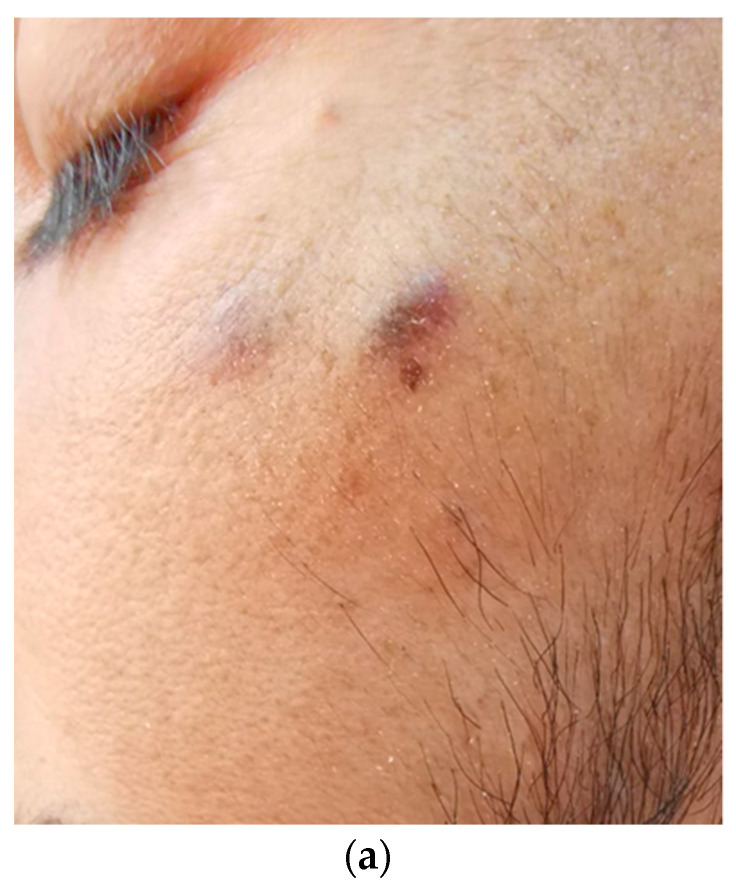
Nodular KS lesions on the face (**a**) and KS purple cutaneous patch in the left axillary region extending to the anterior thorax (**b**) of a 30-year-old heterosexual Caucasian man with HIV, infected since early childhood.

**Figure 2 tropicalmed-09-00047-f002:**
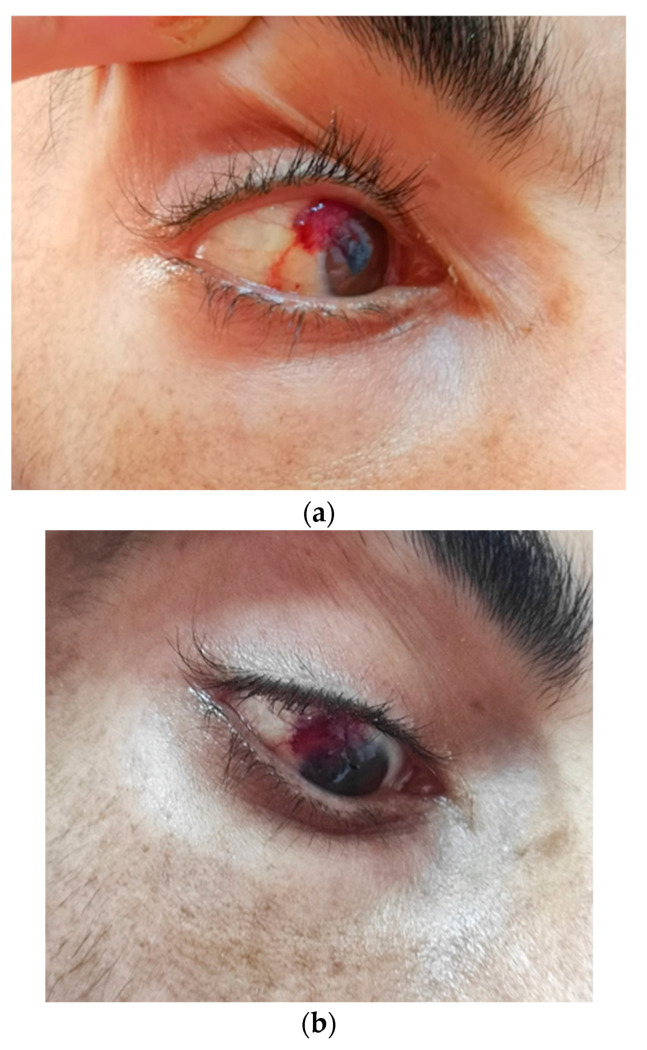
KS red conjunctival nodular lesion in the right eye of a 30-year-old heterosexual Caucasian male, infected with HIV since early childhood, looking up (**a**) and down (**b**).

**Figure 3 tropicalmed-09-00047-f003:**
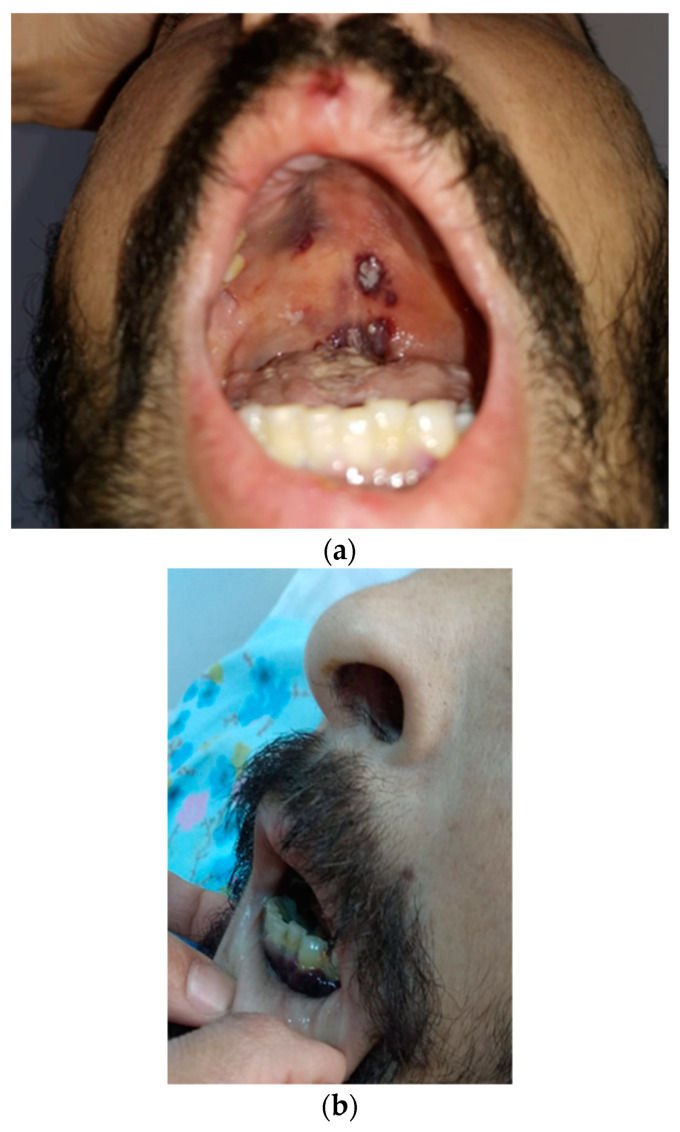
Patches and nodular purple-brown mucosal KS lesions in the gums (**a**) and hard palate (**b**).

**Figure 4 tropicalmed-09-00047-f004:**
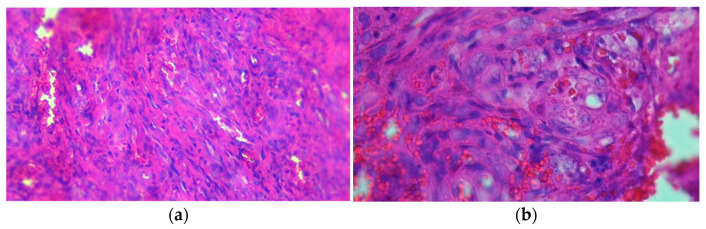
(**a**) Proliferation of spindle cells arranged in intersecting bundles with slit-like vascular spaces, hematoxylin eosin staining (HE) × 200. (**b**) Intracytoplasmic hyaline globules; focal formation of perinuclear vacuoles with red blood cells (autoluminescence), HE × 400.

## Data Availability

All data regarding the findings are available within the manuscript.

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
