# Peer review of "Unusual Localization of AIDS-Related Kaposi’s Sarcoma in a Heterosexual Male during the COVID-19 Pandemic: A Case Report"

_tropicalmed, 2024, doi:10.3390/tropicalmed9020047_

Round 1

Reviewer 1 Report

Comments and Suggestions for Authors

The presented case study addresses a young male with a curious presentation of Kaposi's sarcoma which deteriorated rapidly, ultimately leading to the patient's death only 1 month after the appearance of KS lesions. The patient had a long history of HIV infection and importantly anti-retroviral failure prior to the diagnosis and the authors have available information concerning the long-term health of the patient from HIV diagnosis at the age of 12 until his death. It is unfortunate that KSHV specific immunohistochemistry could not be provided nor KSHV serology or virology, which would have strengthened the case report significantly. That said, the case will likely interest readers of TropicalMed and those in the field of Kaposi's sarcoma and oncogenic viruses. 

The strengths of the case report are the unusual nature of the case as well as the contribution of such a case from a geographic region about which there is scarce information. The authors have put this manuscript forward as a case report and a literature review (stated in the title) but I would rather it be presented as a case report only. The review of the literature included in the paper still has a place in the case report but if submitting a literature review would be lacking. I would rephrase the title to exclude mention of a literature review. 

As a case report, the manuscript is worth publishing. 

Please see below specific comments for the authors: 

Title: I suggest remove "and Review of Literature"

Abstract lines 29 - 30: it is unclear what is meant by "(and other tests)" when only the CD4 count is reported. 

Lines 31 - 33: Rephrase for accuracy based on what is presented in the paper. The authors do not "analyze" the role of co-infections in the course of KS as no actual data on the presence of co-infections is presented. Perhaps "considered" or "discussed" are better phrases. Similarly, think carefully if the COVID-19 pandemic is indeed the cause of delayed diagnosis in this case and rephrase to more accurately depict what is presented. 

Introduction, line 38: typo - remove "intr"

Line 45: double space, check for this throughout.

Line 45: it is unclear what is meant by "more frequent damage". 

Lines 49 - 54: There is now a fifth epidemiological form described. See Vangipuram and Tyring (2019): PMID: 29888407, DOI: 10.1111/ijd.14080.

Lines 58 - 61: Rephrase for clarity. KSHV is the cause of all cases of KS but while it is necessary for KS development is it considered not sufficient, requiring other factors such as immune suppression. 

Line 96: Is the reference to Figure 1 here correct?

Line 112: remove space between "iodophi" and "lic"

Lines 121 -123: Can this be placed chronologically? How long after initial admission did the patient worsen?

Lines 139 - 140: Epidemic KS (related to AIDS) is considered more aggressive than classic KS. It is not necessary to mention "men who have sex with men" in this context. Simply state that the patient is heterosexual and did not report extramarital sexual relationships. 

Literature review: consider rephrasing these sections to directly relate to the case. The discussion is interesting but does not refer to the case. It is important to also include more information on the other KSHV-associated pathologies which can be concurrent with KS. For example, KSHV inflammatory cytokine syndrome is a differential diagnosis that should be considered here. 

Comments on the Quality of English Language

The quality of English language is acceptable but could be improved with language editing. 

Author Response

ANSWER TO COMMENTS AND SUGGESTIONS OF REVIEWER 1

Manuscript: Unusual Localization of AIDS-Related Kaposi Sarcoma in a Heterosexual Male during the COVID-19 Pandemic: A Case Report

Submitted to Tropical Medicine and Infectious Disease

Dear Reviewer 1,

Thank you for your valuable observations. We have revised the muanuscript and answer to you interest points:

R1: Title: I suggest remove "and Review of Literature"

A1: We removed Review of Literature as you suggested.

R2: Abstract lines 29 - 30: it is unclear what is meant by "(and other tests)" when only the CD4 count is reported.

A2: We revised the paragraph.

R3: Lines 31 - 33: Rephrase for accuracy based on what is presented in the paper. The authors do not "analyze" the role of co-infections in the course of KS as no actual data on the presence of co-infections is presented. Perhaps "considered" or "discussed" are better phrases. Similarly, think carefully if the COVID-19 pandemic is indeed the cause of delayed diagnosis in this case and rephrase to more accurately depict what is presented.

A3: We revised the formulation.

R4: Introduction, line 38: typo - remove "intr"

A4: We removed it.

R5: Line 45: double space, check for this throughout.

A5: We remove a space.

R6: Line 45: it is unclear what is meant by "more frequent damage".

A6: We replaced „damage” with involvement, to be more clear.

R7: Lines 49 - 54: There is now a fifth epidemiological form described.

See Vangipuram and Tyring (2019): PMID: 29888407, DOI: 10.1111/ijd.14080.

A7: We completed the classification with the 5th form of KS - nonepidemic KS and we adapted the references, including the study of Vangipuram and Tyring.

R8: Lines 58 - 61: Rephrase for clarity. KSHV is the cause of all cases of KS but while it is necessary for KS development is it considered not sufficient, requiring other factors such as immune suppression.

A8: We rephrase as you suggested ‟Nowadays it is considered that KSHV is the cause of all cases of KS but while it is necessary for KS development is it considered not sufficient, requiring other factors such as immune suppression.”

R: Line 96: Is the reference to Figure 1 here correct?

A8: We replaced „Figure 1” with „Figure A1 and Table A1”.

R9: Line 112: remove space between "iodophi" and "lic"

A9: We removed the space between "iodophi" and "lic".

R10: Lines 121 -123: Can this be placed chronologically? How long after initial admission did the patient worsen?

A10: All the photos were taken on the same day (3rd day of hospitalization), but from different angles and with different lights. The state was severe from the begining, continuing to worse and he died in the 10th day after admittion.

R11: Lines 139 - 140: Epidemic KS (related to AIDS) is considered more aggressive than classic KS. It is not necessary to mention "men who have sex with men" in this context. Simply state that the patient is heterosexual and did not report extramarital sexual relationships.

A11: We rephrase as you suggested ‟ The patient is heterosexual, he  did not reported extramarital sexual relationships, but has developed an aggressive form”.

R12: Literature review: consider rephrasing these sections to directly relate to the case. The discussion is interesting but does not refer to the case. It is important to also include more information on the other KSHV-associated pathologies which can be concurrent with KS. For example, KSHV inflammatory cytokine syndrome is a differential diagnosis that should be considered here.

A12: Concordant with the suggestion of an other reviewer, we have compleated the discussions.

Comments on the Quality of English Language

The quality of English language is acceptable but could be improved with language editing. 

English language will be revised by a native English editor.

Submission Date: 10.02.2024

Reviewer 2 Report

Comments and Suggestions for Authors

Unusual Localization of AIDS-Related Kaposi Sarcoma in a Heterosexual Male during the COVID-19 Pandemic: A Case Report and Review of Literature 

Manuela Arbune et al.

Tropical medicine 

The chronological link with COVID-19. And the onset of diffuse KS in severely immunocompromised HIV persons is of interest and should be the focus of this case report more the unusual site of KS localisation. 

Was the patient infected via transfusion in his childhood?

PCR for HHV-8 can be positive in the blood for extensive KS patient, serology testing is not a validated test, and serology testing must be removed from the text. 

The interplay between HIV, HHV-8 et SARSCOv-2 should be developed and one relevant citation added. 

Patient may have presented with KS along with multicentric Castleman disease due to enlarged lymph nodes et effusion. 

Peng X, . Sharing CD4+ T Cell Loss: When COVID-19 and HIV Collide on Immune System. Front Immunol. 2020 Dec 15;11:596631.

Tajarernmuang P,. Intractable pleural effusion in Kaposi sarcoma following antiretroviral therapy in a Caucasian female infected with HIV. BMJ Case Rep. 2020 Feb 28;13(2):e233335.

Lambarey H,. Reactivation of Kaposi's sarcoma-associated herpesvirus (KSHV) by SARS-CoV-2 in non-hospitalised HIV-infected patients. EBioMedicine. 2024

Discussion:  

In our case, serological or viral level testing for HHV8 was not available, but it is demonstrated that more than 90% of KS cases are associated with this infection.

It is now well established that KS is always associated with HHV-8, Therefore this sentence must be modified.  

Other factors that may have trigger the onset of KS lesions should be cited with reference and descriptions should be very brief. 

Give more explanation why the patient failed or stop having ART as the outcome is so depression for 2024. 

Comments on the Quality of English Language

English and typo should be improved and corrected respectively 

Author Response

ANSWER TO COMMENTS AND SUGGESTIONS OF REVIEWER 2

Manuscript: Unusual Localization of AIDS-Related Kaposi Sarcoma in a Heterosexual Male during the COVID-19 Pandemic: A Case Report

Submitted to Tropical Medicine and Infectious Disease

Dear Reviewer 2,

Thank you for your valuable observations. We have revised the manuscript and answer to you interest points:

R1: Was the patient infected via transfusion in his childhood?

A1: The patient is part of the special Romanian HIV paediatric cohort, that is known because thousands of babies were HIV infected during the communist regime, between 1987-1990, by hospital procedures, involving parenteral treatments, or transfusions multiple. We found in his medical history multiple hospitalization during the first year of life, when the medical procedures were unsafe, due to the treatments with reusable syringes and needles, but we have not found specific documents to notify if he received transfusions.

R2: PCR for HHV-8 can be positive in the blood for extensive KS patient, serology testing is not a validated test, and serology testing must be removed from the text. 

A2: We agree that serology testing is not a validated test, but some studies confirm that the concordance between PCR for HHV8 and serology testing is high.

R3: The interplay between HIV, HHV-8 et SARSCOv-2 should be developed and one relevant citation added. 

A3: We added some citations and developed the interplay between HIV, HHV8 and SarsCov2.

R4: Patient may have presented with KS along with multicentric Castleman disease due to enlarged lymph nodes et effusion. 

Peng X, . Sharing CD4+ T Cell Loss: When COVID-19 and HIV Collide on Immune System. Front Immunol. 2020 Dec 15;11:596631.

Tajarernmuang P,. Intractable pleural effusion in Kaposi sarcoma following antiretroviral therapy in a Caucasian female infeacted with HIV. BMJ Case Rep. 2020 Feb 28;13(2):e233335.

Lambarey H,. Reactivation of Kaposi's sarcoma-associated herpesvirus (KSHV) by SARS-CoV-2 in non-hospitalised HIV-infected patients. EBioMedicine. 2024

A4: We have completed and added new references.

R5: Discussion:  

In our case, serological or viral level testing for HHV8 was not available, but it is demonstrated that more than 90% of KS cases are associated with this infection.

It is now well established that KS is always associated with HHV-8, Therefore this sentence must be modified.  Other factors that may have trigger the onset of KS lesions should be cited with reference and descriptions should be very brief. 

A5: We modified the sentence ‟In our case, serological or viral level testing for HHV8 was not available, but it is demonstrated that KS cases are associated with this infection.”

R6: Give more explanation why the patient failed or stop having ART as the outcome is so depression for 2024. 

A6: The compliance is a very important issue of the patients from the Romanian paediatric cohort. The low formal education, deficient family and social support, HIV-related neurocognitive impairment, are frequently involved in poor compliance. Moreover, living in Romania with HIV in the previous years was perceived as stigma and discrimination, implying the desire of positive person to be accepted by the other people in any way, even hiding the diagnosis or abandoning the medication that could unmask the disease. The therapeutic fatigue due to long time treatment since a very young age, is another very challenging problem for the assistance of this group of patients.

Some of our previous works are illustrative:

https://lumenpublishing.com/journals/index.php/brain/article/view/2394/1993

https://onlinelibrary.wiley.com/doi/epdf/10.7448/IAS.15.6.18251

We have additional comment in the Discussion section, subsection Epidemiological features of HIV infection.
